# Selective culture enrichment and sequencing of feces to enhance detection of antimicrobial resistance genes in third-generation cephalosporin resistant *Enterobacteriaceae*

**Leon Peto**[1,2]*, **Nicola J. Fawcett**[1,2], **Derrick W. Crook**[1,2,3], **Tim E. A. Peto**[1,2], **Martin J. Llewelyn**[4,5]ᵒ, **A. Sarah Walker**[1,2]ᵒ

**1** National Institute for Health Research (NIHR) Health Protection Research Unit on Healthcare Associated Infections and Antimicrobial Resistance, John Radcliffe Hospital, Oxford, England, United Kingdom, **2** Nuffield Department of Medicine, University of Oxford, Oxford, England, United Kingdom, **3** National Infection Service, Public Health England, Colindale, London, England, United Kingdom, **4** Department of Global Health and Infection, Brighton and Sussex Medical School, Falmer, Sussex, England, United Kingdom, **5** Department of Microbiology and Infection, Brighton and Sussex University Hospitals NHS Trust, Brighton, England, United Kingdom

ᴑ These authors contributed equally to this work.
* leon.peto@ndm.ox.ac.uk

**Data Availability Statement:** Sequence data are available from the European Nucleotide Archive, Project Accession Number PRJEB34966.

## Abstract

Metagenomic sequencing of fecal DNA can usefully characterise an individual's intestinal resistome but is limited by its inability to detect important pathogens that may be present at low abundance, such as carbapenemase or extended-spectrum beta-lactamase producing *Enterobacteriaceae*. Here we aimed to develop a hybrid protocol to improve detection of resistance genes in *Enterobacteriaceae* by using a short period of culture enrichment prior to sequencing of DNA extracted directly from the enriched sample. Volunteer feces were spiked with carbapenemase-producing *Enterobacteriaceae* and incubated in selective broth culture for 6 hours before sequencing. Different DNA extraction methods were compared, including a plasmid extraction protocol to increase the detection of plasmid-associated resistance genes. Although enrichment prior to sequencing increased the detection of carbapenemase genes, the differing growth characteristics of the spike organisms precluded accurate quantification of their concentration prior to culture. Plasmid extraction increased detection of resistance genes present on plasmids, but the effects were heterogeneous and dependent on plasmid size. Our results demonstrate methods of improving the limit of detection of selected resistance mechanisms in a fecal resistome assay, but they also highlight the difficulties in using these techniques for accurate quantification and should inform future efforts to achieve this goal.

**Funding:** The research was funded by the National Institute for Health Research Health Protection Research Unit (NIHR HPRU) in Healthcare Associated Infections and Antimicrobial Resistance at the University of Oxford in partnership with Public Health England (PHE) [HPRU-2012-10041], the NIHR Oxford Biomedical Research Centre, and a Medical Research Council UK Clinical Research Training Fellowship to NJF. The views expressed are those of the author(s) and not necessarily those of the NHS, the NIHR, the Department of Health or PHE. DWC and TEAP are NIHR Senior Investigators. The funder had no role in study design, data collection and analysis, decision to publish, or preparation of the manuscript.

**Competing interests:** The authors have declared that no competing interests exist.

## Introduction

The intestinal microbiome is one of the most important reservoirs of clinically relevant antimicrobial resistance (AMR) genes [1]. Accurate detection and quantification of AMR genes within an individual's gut microbiome (their gut 'resistome') could allow the impact of different types of antibiotic exposures to be evaluated and guide interventions to reduce AMR. This would be particularly helpful for antimicrobial stewardship, as current policies often categorise agents by their clinical spectrum alone, which may poorly reflect their actual impact on the gut microbiome and resistome [2]. Reliable quantification is an important feature of a resistome assay, as although simply classifying individuals as 'colonized' or 'not colonized' is useful in many settings, often the selection of resistance consists of increasing the abundance of organisms already present [3,4]. In addition, the abundance of antimicrobial resistant organisms in the gut has been associated with risk of invasive infection and onward transmission [3,5].

Unfortunately, all current methods of measuring the fecal resistome have major limitations. Quantitative culture requires further genotyping to define resistance elements and can only practically be used for a small number of species present in feces [6], and targeted molecular approaches, such as qPCR, cannot simultaneously detect the dozens of AMR genes and mutations that may be present in a sample from among thousands of known types [7,8]. Because of these shortcomings, metagenomic sequencing performed directly from extracted fecal DNA is increasingly being used to characterise the gut resistome, as it produces millions of short DNA reads that can be matched to a catalogue of thousands of AMR genes, theoretically producing a representative, quantitative description of AMR genes in a sample [1,7,9].

However, a major limitation of direct sequencing is the inability to detect clinically important resistance genes present at low abundance [10]. In particular, the *Enterobacteriaceae*, which include major drug resistant human pathogens such as *Escherichia coli*, *Salmonella* and *Klebsiella* species, may be present in feces at levels too low for AMR gene detection by direct sequencing. In one recent study, metagenomic sequencing detected an extended spectrum beta-lactamase (ESBL) in just 12 of 26 (46%) culture positive stool samples, missing all 9 samples with an abundance of $<10^6$ CFU/g [11]. Increasing the depth of sequencing to detect scarce organisms seems appealing but is not currently feasible. For example, it would not be unusual for a resistant *E. coli* to be present at an abundance of $10^3$ cells/g in feces with a total organism count of $10^{11}$ cells/g [11,12], so using short read sequencing to detect a 1kb AMR gene in this organism would require hundreds of billions of reads, costing many hundreds of thousands of dollars.

Here we assess whether a short period of selective culture enrichment of fecal samples before sequencing can improve the detection and quantification of AMR in *Enterobacteriaceae* by raising their abundance above the threshold of detection. We show that the methods tested can lower the limit of detection of selected resistance mechanisms, but that they are unable to provide reliable quantification, highlighting the difficulties of using these, or other similar approaches.

## Materials and methods

### Experiment 1—Comparison of DNA extraction methods

Preliminary experiments were performed to identify a DNA extraction protocol that would efficiently extract resistance genes present in *Enterobacteriaceae*. As acquired resistance genes in *Enterobacteriaceae* are typically present on plasmids, we hypothesised that a plasmid extraction protocol would increase detection of these genes. The QuickLyse Miniprep Kit (Qiagen), a spin column-based plasmid extraction kit, was compared to a standard fecal DNA extraction

protocol, the QIAamp Fast DNA Stool Mini Kit (Qiagen) modified to include mechanical lysis with bead beating. One hundred milligram aliquots from two different volunteer fecal sample were spiked with one of two *Enterobacteriaceae* carrying plasmid-associated beta-lactamases (*K. pneumoniae* with NDM-1 and CTX-M-15, or *E. cloacae* with KPC-2 and TEM-1). Spike organisms were added at a concentration of $10^8$ CFU/g and negative controls with no spike were run in parallel. Following DNA extraction using the two methods, sequencing was performed on Illumina MiSeq with a mean sequencing depth of 1.4 million paired reads per sample.

## Experiment 2 –Establishing the optimal duration of enrichment culture

Before assessing a protocol of enrichment-culture and direct sequencing, we first had to determine the optimal duration of culture. We therefore performed an experiment in which 100mg of a fecal sample was incubated in Mueller-Hinton broth containing metronidazole (20mg/L) and vancomycin (20mg/L), repeated in duplicate. These antibiotics were added to suppress the growth of anaerobic and gram-positive organisms but allow growth of *Proteobacteria*, which would primarily consist of *Enterobacteriaceae*. Quantitative culture was performed after 0, 4, 6, 8, 10 and 24 hours incubation.

## Experiment 3 –Sequencing after selective enrichment with vancomycin and metronidazole

We then assessed the effect of this selective enrichment protocol on sequencing-based detection of AMR genes present in *Enterobacteriaceae*. One hundred milligram aliquots from a single fecal sample were spiked with *E. coli* containing NDM-1 at concentrations of $0 – 5\text{x}10^5$ CFU/g, repeated in triplicate. These had 6 hours of enrichment culture in broth containing metronidazole and vancomycin as above, followed by DNA extraction from the pelleted broth with the Fast DNA Stool kit and sequencing on Illumina MiSeq with a mean sequencing depth of 1.7 million paired reads per sample.

## Experiment 4 –Sequencing after selective enrichment with vancomycin, metronidazole and cefpodoxime

A second culture-enrichment protocol was tested, which was the same as above apart from the addition of cefpodoxime (1mg/L) and use of the QuickLyse Miniprep Kit for DNA extraction. To establish the performance of this protocol, another spiking experiment was performed using fecal samples from two volunteers. One hundred milligram aliquots from the samples were spiked with one of two different carbapenemase-producing *Enterobacteriaceae* (CPE) (NDM-1 *K. pneumoniae* or KPC-2 *E. cloacae*) at spikes ranging from $0 – 5\text{x}10^7$ CFU/g. Unenriched controls were run in parallel and all conditions were repeated in duplicate. In addition, to allow estimation of the number of CPE organisms in post-enrichment samples using sequencing data, a standard spike of $10^7$ CFU of *Staphylococcus aureus* was added to all samples except negative controls immediately prior to DNA extraction. Sequencing was performed on Illumina MiSeq to a mean depth of 1.4 million paired reads per sample. Quantitative culture was also performed on all samples.

## Specimens

The Antibiotic Resistance in the Microbiome Oxford (ARMORD) study gathers fecal samples from participants to study antimicrobial resistance. The study was approved by Leicester Research Ethics Committee (reference 15/EM/0270) and all participants provided informed

consent. Stool samples from two ARMORD participants were used for experiments 1, 3 and 4. Participant A was a hospital inpatient with exposure to a third-generation cephalosporin antibiotic (ceftazidime) 48 hours previously, participant B was a healthy volunteer with no recent antibiotic exposure. Samples were frozen at -80°C within 2 hours of collection and were confirmed to be culture-negative for ESBL producing *Enterobacteriaceae* using overnight growth in Mueller Hinton broth containing 1mg/L cefpodoxime (Sigma-Aldrich) followed by inoculation onto ESBL Brilliance agar (Oxoid) for 24 hours at 37°C in aerobic conditions [13]. Experiment 2 used a discarded fecal sample from the clinical microbiology laboratory at the Oxford University Hospitals NHS Foundation Trust.

## Quantitative culture

Modified track plating was used to quantify *Enterobacteriaceae* in fecal samples [14]. Samples were serially diluted 10-fold in peptone water with pipette mixing at each step. Culture of 10μl from dilutions 1:10–1:10$^8$ was performed on CHROMagar Orientation (BD) and ESBL Brilliance agar (Oxoid). Colonies were counted after incubation at 37°C for 16 hours in aerobic conditions.

## Experimental bacterial strains

Three CPE strains were used for the spiking experiments described above, grown on CRE Brilliance agar (Fisher Scientific) at 37°C in aerobic conditions.

1. *E. coli* H17. Contains NDM-1 beta-lactamase on unresolved plasmid, isolated from a hospital outbreak in Nepal (SAMN02885348) [15].

2. *Klebsiella pneumoniae* PMK3. Contains NDM-1 and CTX-M-15 beta-lactamases on a 305kb plasmid, present at an estimated 1.8 copies per chromosome. Isolated from a hospital outbreak in Nepal (SAMN02885389) [15].

3. *Enterobacter cloacae* CAV1668. Contains KPC-2 and TEM-1 beta-lactamases on a 43kb plasmid, present at an estimated 1.9 copies per chromosome. Isolated from a hospital outbreak in Virginia, USA (SAMN03733827) [16].

MRSA252 (SAMEA1705935) was used for the standard *S. aureus* spike, which was grown on blood agar (Fisher Scientific) at 37°C in aerobic conditions.

## Spiking

Overnight growth of bacteria was used to make a suspension of 5 MacFarland in nutrient broth with 10% glycerol (NBG) (Oxoid, BO1297G), measured using a BD PhoenixSpec nephelometer. Serial 10-fold dilutions were frozen at -80°C with culture from thawed aliquots used to define concentrations. Stool samples were vortexed with molecular water (Fisher Scientific) on ice at a ratio of 1:2 to create a pipettable suspension, 300mg of which was used per set of experimental conditions.

## Enrichment

Spiked samples were added to 20ml Mueller-Hinton broth (Sigma-Aldrich) containing 20mg/L vancomycin, 20mg/L metronidazole, and for experiment 4, 1mg/L cefpodoxime (all antibiotics Sigma-Aldrich). These were incubated at 37°C and shaken at 190rpm for 6 hours, then, for sequencing experiments, immediately centrifuged at 3200g for 10 min at 4°C to pellet bacteria. Supernatant was discarded and the pellet resuspended in 1ml molecular water.

## DNA extraction

Two DNA extraction protocols were used.

1. QuickLyse Miniprep Kit (Qiagen), used in experiments 1 and 4. Samples were centrifuged at 100g for 5 minutes to remove large particles, and DNA was then extracted from the supernatant according to the manufacturer's instructions. Briefly, bacterial cells were pelleted by centrifugation at 17,000g for 2 minutes and resuspended in 400μl ice-cold complete lysis solution by vortexing for 30s. After 3 minutes at room temperature the lysate was transferred to a QuickLyse spin column and centrifuged at 17,000g for 60s. The spin column was washed with 400μl QuickLyse wash buffer and centrifuged at 17,000g for 60s. After discarding the flow-through, the columns were dried by centrifuging at 17,000g for 60s. DNA was eluted from the spin column in 50 μl QuickLyse elution buffer by spinning at 17,000g for 60s.

2. QIAamp Fast DNA Stool Mini Kit (Qiagen) plus mechanical lysis with bead beating, used in experiments 1 and 3. Samples were added to 1ml Stool Transport and Recovery buffer (Roche) in a 2ml Lysing Matrix E tube (MP Biomedicals), followed by bead beating for 2 x 40s with a FastPrep-24 5G instrument (MP Biomedicals) and incubation at 95˚C for 5 minutes. Samples were then centrifuged at 1,000g for 60s. Supernatant was transferred to 1ml InhibitEX buffer, after which the manufacturer's instructions were followed for bacterial DNA extraction. Briefly, this involved mixing with 15μl proteinase K, 400μl AL lysis buffer and 400μl 100% ethanol. This was transferred to a QIAamp spin column and washed with 500μl AW1 and 500 μl AW2 buffers. DNA was then eluted in 50μl molecular water.

## DNA library preparation and sequencing

Extracted DNA was purified using Ampure XP beads (Agilent). Fifty microliters of DNA was vortexed with 90μl Ampure XP beads and washed twice with 80% ethanol before elution in 25μl molecular water. DNA was normalized to 0.2ng/μl and libraries prepared from 1μg DNA using the Nextera XT Library preparation kit (Illumina) according to the manufacturer's instructions. Normalization and QC of libraries was performed using a Qubit 2.0 fluorometer (Thermo Fisher) and 2200 Tapestation (Agilent). Paired-end 300bp sequencing was performed on an Illumina MiSeq, with 10–16 samples multiplexed per sequencing run.

## Bioinformatics

Human DNA reads (representing <4% of sequences in all samples) were identified using the Kraken taxonomic classifier and removed prior to analysis [17]. Quality control metrics were assessed with FastQC [18], and BBduk was used to remove Illumina adapters and quality-trim reads using a phred-score cut-off of 10 [19]. MetaPhlAn2 was used for taxonomic classification [20], except for normalization to *S. aureus*, where Kraken was used to classify reads to the relevant species. BBMap was used to map reads to beta-lactamase genes, *gyrA* and plasmids using the relevant genes from assembled genomes specified in bacterial strains as the reference [19]. Plasmid copy number was estimated by mapping reads from sequenced isolates to a closely related closed reference genome using BWA-MEM [21]. For PMK3, the PMK1 reference was used (CP008929-CP008933), and for CAV1668, the CAV1668 reference was used (CP011582-CP011584). SAMtools was used to filter out supplementary alignments and calculate depth of coverage [22].

## Normalization to *S. aureus*

To allow quantification of CPE spike organisms after enrichment in experiment 4, a standard inoculum of $10^7$ CFU *S. aureus* was added to samples after enrichment and immediately prior to DNA extraction. Detection of DNA reads from different organisms depends on several factors such as genome size, efficiency of DNA extraction, efficiency of sequencing and efficiency of bioinformatic classification. These can be combined into a single measure of relative detection efficiency (RDE) compared to *S. aureus*, which was established using unenriched samples in which the concentration of the spike organism and *S. aureus* were both known. For each CPE spike organism, the mean RDE was determined from the 2 unenriched samples in experiment 4 spiked with $5x10^7$ CFU/g CPE and $10^7$ CFU *S. aureus*. The following formula was used, based on the relative proportions of the known CPE and *S. aureus* CFU, and the relative proportions of the reads assigned to the CPE species and *S. aureus* by Kraken:

$$\frac{Spike\ organism\ reads}{S.\ aureus\ reads} = \text{RDE x } \frac{Spike\ CFU}{S.\ aureus\ CFU}$$

The calculated RDE was then used to estimate the number of CFUs present in samples after enrichment.

## Statistical analysis

All statistical analyses were performed in R, version 3.5.0. Means were compared using a Welch two sample t-test, which allows unequal variances in the two groups.

## Results

### Experiment 1 —Comparison of DNA extraction methods

Use of the QuickLyse plasmid extraction method increased the number of reads mapping to plasmid-associated beta-lactamases to a mean of 15.7 Reads per Kb per million (RPKM) (95% CI 6.2–25.1) compared to a mean of 4.4 RPKM (95% CI 2.4–6.4, t-test for difference p = 0.03) using the Fast DNA Stool kit (S1 Fig).

### Experiment 2 –Establishing the optimal duration of enrichment culture

Selective enrichment of *Enterobacteriaceae* in broth containing vancomycin and metronidazole demonstrated that they were in log-phase growth up to approximately 8 hours of incubation (Fig 1). Six hours was selected as the optimal incubation period, as this provided approximately $10^4$-fold enrichment while preserving the ratios of the different *E. coli* strains present at baseline.

### Experiment 3 –Sequencing after selective enrichment with vancomycin and metronidazole

Culture-enrichment in broth containing vancomycin and metronidazole showed that at all spiking concentrations tested, the spike organism was barely detectable above endogenous *E. coli* present in the sample, with a mean relative abundance of *E. coli* of 3.5% (range 2.9–4.4%) in 16 unenriched samples, and 98.0% (range 97.2–98.6%) in 16 enriched samples (S2 Fig). The NDM-1 gene was scarcely measurable in any sample, being detected in just a single read in 3 enriched samples: 2 of 3 spiked at $5x10^5$ CFU/g, and 1 of 3 spiked at $5x10^4$ CFU/g, in each case corresponding to an abundance of <1 RPKM (starred samples in S2 Fig).

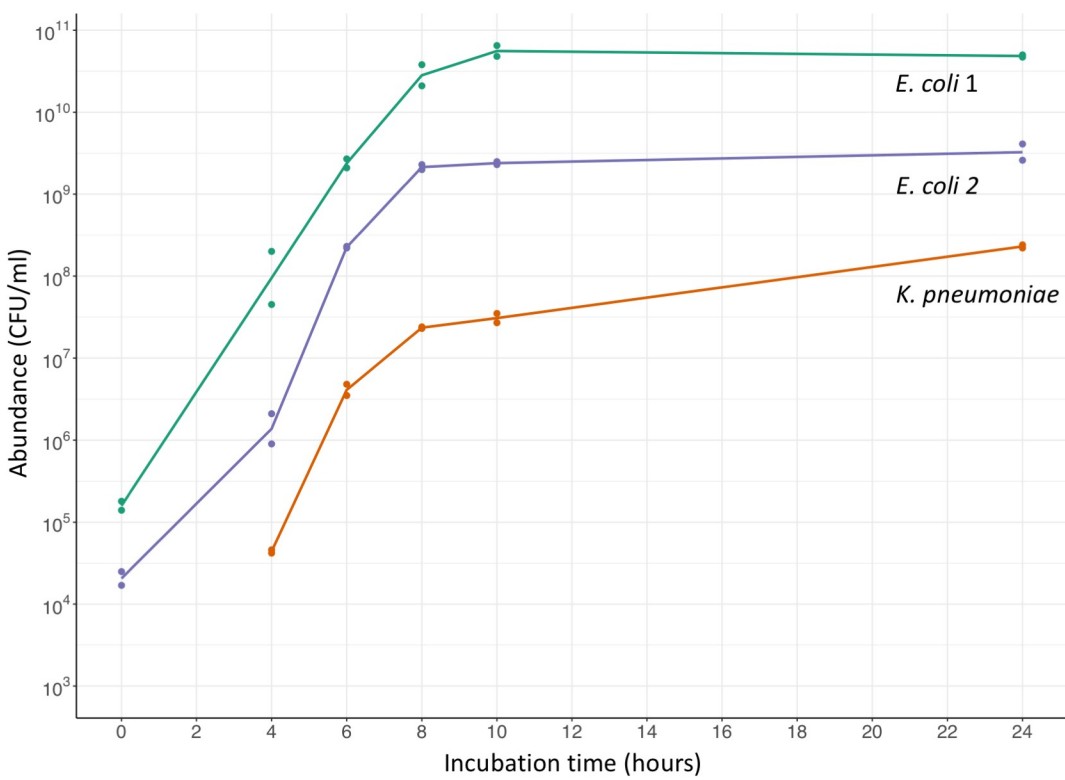

**Fig 1. Growth of *Enterobacteriaceae* during incubation of a fecal sample.** A single sample was incubated in broth containing vancomycin and metronidazole. Two distinct morphotypes of *E. coli* were present. No *K. pneumoniae* were isolated at 0 hours, implying presence below the limit of detection of $10^3$ CFU/g. Lines join the geometric means of 2 replicates.

## Experiment 4 —Sequencing after selective enrichment with vancomycin, metronidazole and cefpodoxime

As enrichment with metronidazole and vancomycin alone did not allow detection of a resistant subpopulation of *Enterobacteriaceae*, a revised protocol was developed which added cefpodoxime to suppress growth of susceptible *Enterobacteriaceae* and used the QuickLyse plasmid extraction method from experiment 1.

Using this revised protocol, CPE spikes above $5x10^5$ CFU/g led to a relative abundance of $\geq$90% the spike organism after enrichment, based on taxonomic classification of sequence data (Fig 2). At lower spiking concentrations, there was a marked difference between the spike organisms. At an initial spike of $5x10^2$ CFU/g, *K. pneumoniae* had a mean relative abundance of 58% compared to 3% for *E. cloacae*. In all unenriched samples the spike organism was present at a relative abundance of <10%.

As our objective was to measure the absolute, as well as relative, abundance of organisms present in samples, we estimated the abundance of spike organisms after enrichment by comparing reads assigned to the spike to those assigned to the *S. aureus* standard. Abundance estimation in the enriched samples showed that the number of spike organisms increased with higher initial spiking concentration even after the relative abundance had plateaued (Fig 3). Comparing these sequencing-based estimates to measurement by quantitative culture showed reasonable correlation at abundances below $10^9$ CFU, corresponding to initial spikes <$5x10^5$ CFU/g, with all estimates within 1 $\log_{10}$ of the culture value (Fig 4). However, at higher

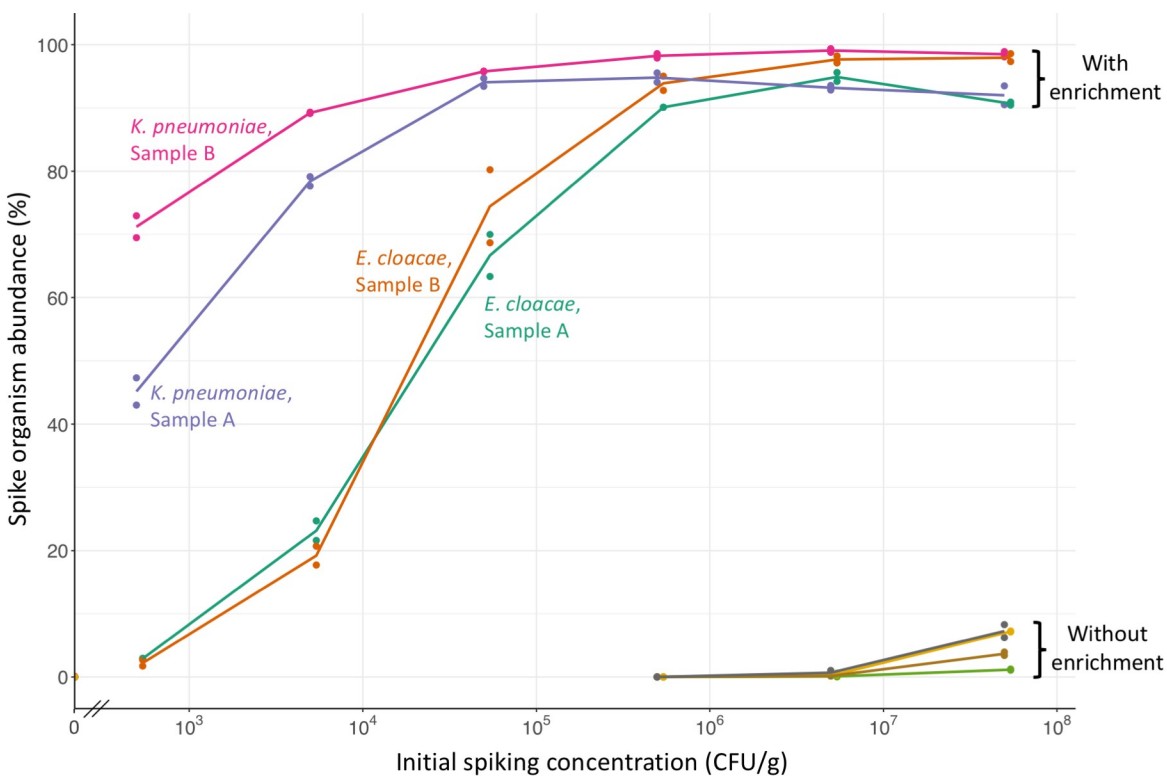

**Fig 2. Relative abundance of CPE following culture-enrichment of spiked fecal samples.** Initial CPE spike versus measured abundance of spike organism by sequencing, with or without enrichment in broth containing cefpodoxime, vancomycin, and metronidazole. Two spike CPE (*K. pneumoniae* & *E. cloacae*) and two volunteer samples (A & B) were used. Lines join the arithmetic means of 2 replicates. Unspiked samples are shown on the y-axis.

abundances, where the number of spike CFU in the sample was more than 300-fold higher than the number of *S. aureus* CFU, the number of spike organisms estimated by sequencing consistently underestimated culture values.

Detection of the carbapenemase genes differed markedly between the two spike organisms (solid lines in Fig 5). The KPC-2 gene present in *E. cloacae* was detected at a mean 8 RPKM (range 4–12) at the lowest spike of $5 \times 10^2$ CFU/g, implying a limit of detection of approximately 50 CFU/g per million reads. In contrast, the NDM-1 gene present on *K. pneumoniae* was detected at around 100 RPKM even at the lowest initial spike and hardly changed with higher spiking concentrations, in keeping with the higher abundance of *K. pneumoniae* noted above. This prevented estimation of the limit of detection of NDM-1 in this context.

At the highest spiking concentrations, the number of RPKM mapping to the carbapenemase gene was around 4 times higher in *E. cloacae* than *K. pneumoniae*, even though both organisms were present at >90% abundance. This difference did not appear to relate to the plasmid copy number, as in sequenced isolates the carbapenemase genes are present at similar copy numbers of 1.8–1.9. To explore the possibility that this difference related to efficiency of plasmid extraction, we compared the number of reads mapping to the plasmid-associated carbapenemase to those mapping to the chromosomal *gyrA* gene (dashed lines in Fig 5). In *E. cloacae*, DNA from the 43kb KPC plasmid was extracted more efficiently than the chromosomal *gyrA* gene, leading to greater detection of KPC-2 vs gyrA. In contrast, in *K. pneumoniae* DNA from the 305kb NDM-1 plasmid and *gyrA* was extracted at similar efficiencies.

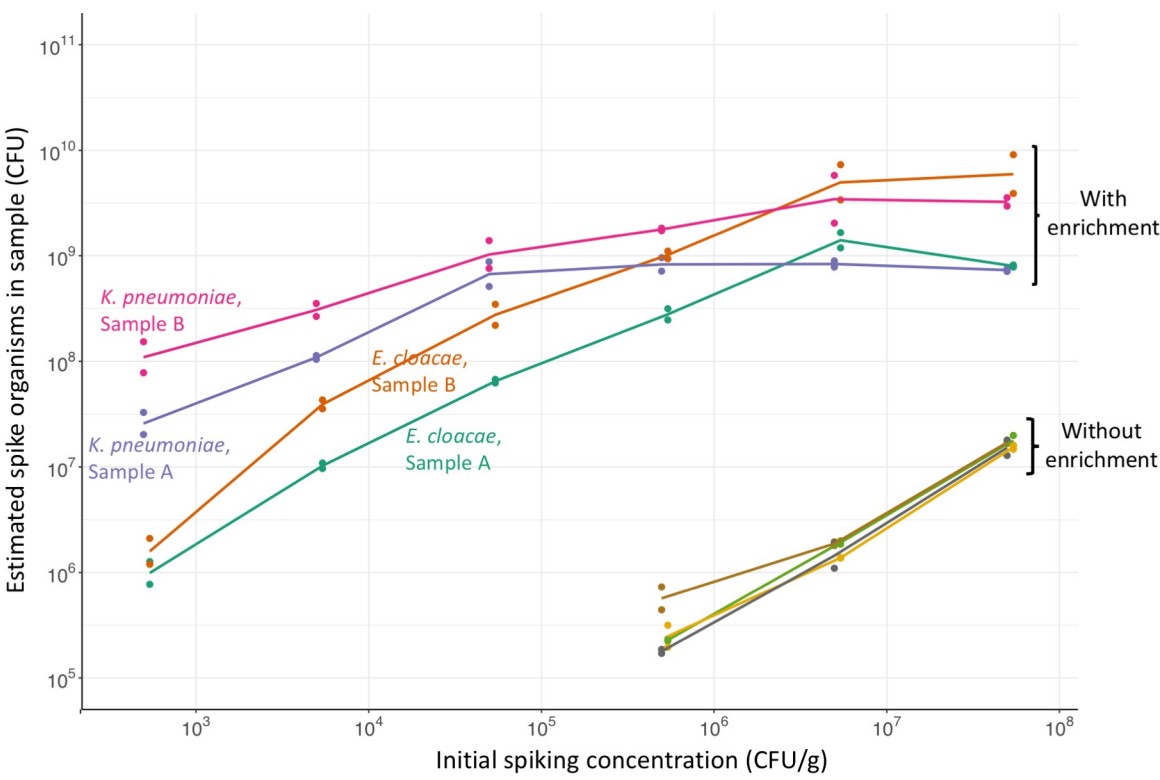

**Fig 3. Estimated number of CPE organisms following culture-enrichment of spiked fecal samples.** Initial CPE spike versus final number of CPE spike organisms estimated from sequence data by normalization to *S. aureus*. With and without enrichment in broth containing cefpodoxime, vancomycin, and metronidazole. Two spike organisms (*K. pneumoniae* & *E. cloacae*) and two volunteer samples (A & B) were used. Lines join the geometric means of 2 replicates.

As well as the NDM-1 plasmid, the *K. pneumoniae* spike also contained DNA mapping to two small plasmids, pEC34A (3.8kb) and pRGRH1815 (7.1kb), although the structure of these in the spike organism could not be resolved with available sequence data [23]. This DNA was extracted at a very high efficiency in enriched samples, and the extraction efficiency relative to chromosomal DNA increased with higher spiking concentrations (S3 Fig). At the highest spikes this small plasmid DNA made up around a quarter of all reads, causing a relative decrease in other *K. pneumoniae* DNA.

## Discussion

Selective culture-enrichment of stool samples has been successfully used by Raymond et al to characterize low abundance constituents of the microbiome that would have been undetectable with standard metagenomic sequencing, albeit without any attempt at quantification [24]. In this study, we aimed to create a simple, quantitative assay of resistance genes in culturable organisms using a short period of selective culture followed by sequencing directly from the enriched sample. We chose culture conditions that would amplify a resistant subpopulation of *Enterobacteriaceae* of particular clinical importance, specifically those resistant to third generation cephalosporins, with the aim of bringing their resistance genes above the threshold of detection. It is important that enrichment of resistant organisms is not at the expense of quantification, so after initial experiments to define the growth of *Enterobacteriaceae* in selective broth culture we chose an incubation period of 6 hours, which was found to produce a $10^4$-

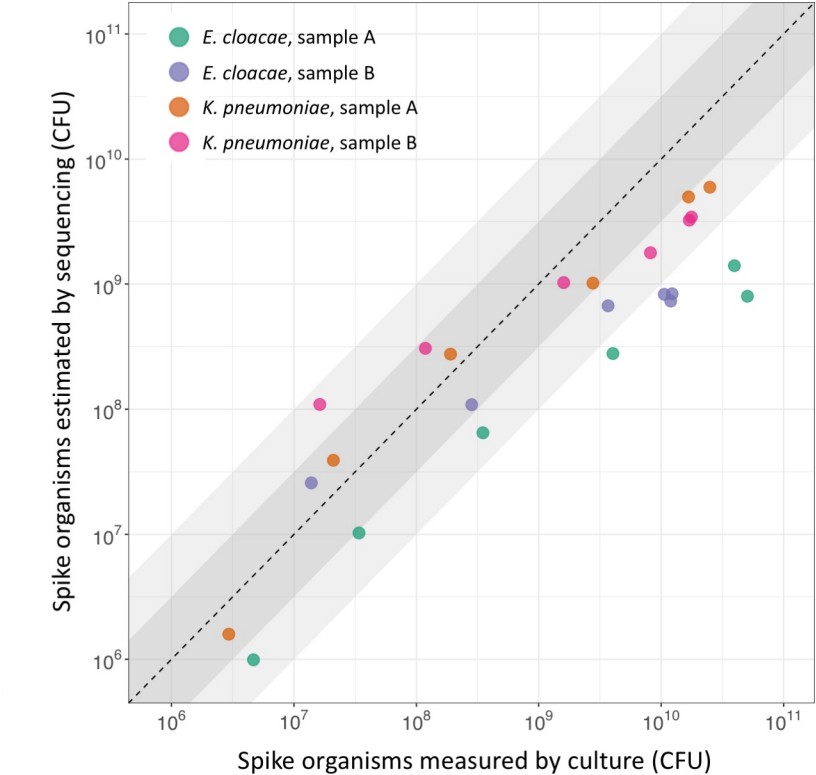

**Fig 4. Comparison of CPE organisms estimated by sequencing and culture.** Number of spike organisms estimated by sequencing compared to measurement by culture. Samples enriched in broth containing cefpodoxime, vancomycin, and metronidazole. Two spike organisms (*K. pneumoniae* & *E. cloacae*) and two volunteer samples (A & B) used. Points represent the geometric means of 2 replicates, shaded areas are within 0.5 and 1.0 $\log_{10}$ of the line of identity.

fold increase without going past log-phase growth, which would have prevented estimation of the starting concentration.

Performance of the enrichment-sequencing protocol was assessed by spiking fecal samples with known concentrations of CPE. While there were minimal differences between samples from different individuals, there were large differences between the two spike organisms. Consistency between the fecal samples, and between replicates, implies that this relates to intrinsic growth characteristics of the strains used in this experiment. Our finding of such large variation in growth between different strains of *Enterobacteriaceae* precludes accurate estimation of their starting concentration unless their growth characteristics are known, undermining one of the aims of the assay. In fact, relatively small differences in growth rates could produce such differences after 6 hours. For example, if two *Enterobacteriaceae* strains had doubling times of 18 minutes and 22 minutes, then this would lead to a 12-fold difference in their abundance after 6 hours incubation. As well as intrinsic differences between bacterial strains, variation in growth rate can also be influenced by total bacterial concentration [25] and the fecal environment [26].

To estimate the absolute abundance of organisms in the sample using sequence data it is necessary to use some form of standard. Although our use of a *S. aureus* standard allowed reasonable estimation of the number of post-enrichment spike organisms at low abundances, it correlated poorly with quantitative culture at higher abundances (Fig 4). This is likely to be related to small amounts of contaminating DNA assigned to the standard, as even in negative controls with no fecal sample a small number of reads were assigned to *S. aureus*. This could

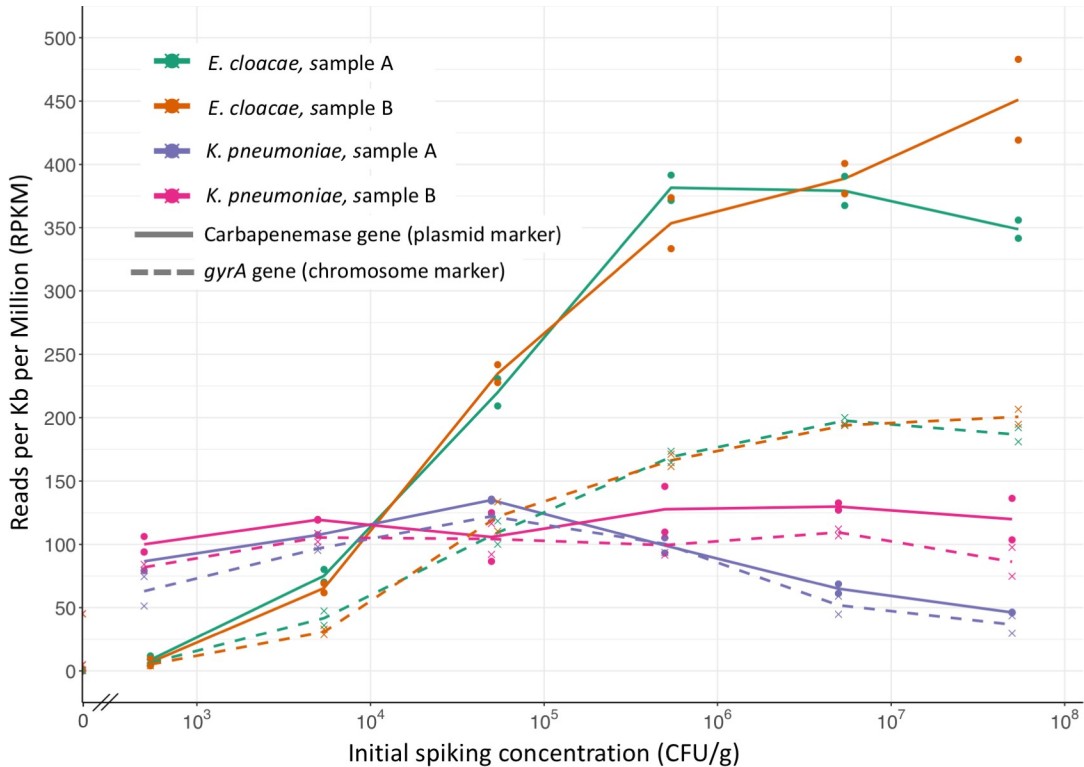

**Fig 5. Detection of genes present in CPE following culture-enrichment of spiked fecal samples.** Initial CPE spike versus RPKM mapping to carbapenemase genes (solid lines, NDM-1 in *K. pneumoniae*, KPC-2 in *E. cloacae*) and gyrA (dashed lines). Samples enriched in broth containing cefpodoxime, vancomycin, and metronidazole. Two spike organisms (*K. pneumoniae* & *E. cloacae*) and two volunteer samples (A & B) were used. Lines join the arithmetic means of 2 replicates. Unspiked samples are shown on the y-axis.

possibly be improved by using an alternative standard with less homology to any possible contaminants or organisms present in the sample.

Another novel aspect of this protocol was the use of plasmid DNA extraction, which increased the detection of plasmid-mediated resistance genes in *Enterobacteriaceae* in preliminary spiking experiments. To our knowledge this is the first study to compare plasmid DNA extraction to standard DNA extraction for resistome assessment in humans. One study assessing AMR genes in environmental metagenomes found that plasmid extraction could increase detection of some resistance genes 37 fold [27], but in our study the effect on the genes of interest was much more modest, with increases of only 3–4 fold. More problematically, plasmid DNA extraction produced unwanted artefacts because of differential extraction efficiency depending on plasmid size. This was noticeable when comparing the 305kb and 43kb carbapenemase containing plasmids, but was much more marked in relation to reads mapping to small 4-7kb plasmids, which were extracted at extremely high efficiency compared to other DNA.

This study has several limitations. It was only possible to assess a limited number of conditions, and the quantitative findings of this study may have been different if using different spike organisms, culture conditions or plasmid extraction methods, although it seems unlikely that the qualitative conclusions would differ. The use of a plasmid extraction kit in some experiments means that measured bacterial abundances will be biased towards organisms that are extracted more efficiently with this method, but this effect should be consistent between

conditions in the same experiment. Finally, although we detected DNA mapping to known small plasmids in samples spiked with *K. pneumoniae*, we are unable to resolve the exact plasmid structures or sizes with available sequencing data.

Given the difficulties with the methods tested here, alternative ways of quantifying scarce AMR genes are needed. A promising alternative is to use target capture, in which a custom library of nucleic acid probes is used to enrich resistance genes directly from a standard metagenomic DNA extract. Using such a method, Noyes et al [28] were able to increase the proportion of sequenced reads mapping to resistance genes over 100-fold, and detected important resistance mechanisms, such as KPC, that were missed with standard metagenomic sequencing. Their workflow also incorporated unique molecular indices to improve the reliability of quantification. Using a different target capture system, Lanza et al increased the proportion of reads mapping to resistance genes 279-fold [10]. Both studies compared target capture and sequencing to standard metagenomic sequencing, but neither were tested on samples with a known concentration of antimicrobial resistant organism or on negative controls. Our study demonstrates the need to validate methods in well characterized conditions before they can reliably be used to make quantitative comparisons.

## Conclusions

Our study has assessed novel approaches to lowering the limit of detection of AMR genes in *Enterobacteriaceae* in a sequencing-based assay of the intestinal resistome. By comparing different strains of *Enterobacteriaceae* we have demonstrated the difficulty of using culture enrichment whilst maintaining quantification. Assessing the differential effect of plasmid DNA extraction by plasmid size has highlighted the artefacts that make it inappropriate for use in a resistome assay requiring accurate quantification. Further attempts at quantitative resistome assessment must either use different methods to amplify scarce AMR genes, or accept the likelihood of missing genes in low abundance organisms, which may include important pathogens such as *Enterobacteriaceae*.

## Supporting information

**S1 Fig. Effect of plasmid extraction on detection of beta-lactamase genes.** Reads mapping to plasmid-associated beta-lactamases in DNA extracted from fecal samples spiked with $10^8$ CPE/g. Two DNA extraction methods were compared using one of two spike CPE (*K. pneumoniae* & *E. cloacae*) and one of two volunteer samples (A & B). QuickLyse extractions were performed in duplicate.
(TIF)

**S2 Fig. *E. coli* abundance following culture-enrichment of spiked feces.** Aliquots of a single fecal sample spiked with NDM-1 *E. coli*, with or without 6 hours of enrichment in broth containing vancomycin and metronidazole. Note that the sample contained endogenous non-CPE *E. coli*. Stars indicate the three samples in which a single DNA read mapped to the NDM-1 gene, with no reads mapping to the gene in any other samples. Lines join the arithmetic means of 3 replicates. Unspiked samples are shown on the y-axis.
(TIF)

**S3 Fig. Detection of plasmid markers present in CPE following culture-enrichment of spiked fecal samples.** Initial *K. pneumoniae* spike versus ratio of plasmid-marker to chromosomal gyrA reads. Samples enriched in broth containing cefpodoxime, vancomycin, and metronidazole. Two volunteer samples (A & B) were used. Lines join the geometric means of 2

replicates.
(TIF)

## Acknowledgments

Carbapenemase-producing *Enterobacteriaceae* strains provided by Nicole Stoesser. Carbapenemase copy estimation performed by Anna Sheppard.

## Author Contributions

**Conceptualization:** Leon Peto, Nicola J. Fawcett, Derrick W. Crook, Tim E. A. Peto, Martin J. Llewelyn, A. Sarah Walker.

**Formal analysis:** Leon Peto.

**Investigation:** Leon Peto, Nicola J. Fawcett.

**Supervision:** Derrick W. Crook, Tim E. A. Peto, Martin J. Llewelyn, A. Sarah Walker.

**Writing – original draft:** Leon Peto.

**Writing – review & editing:** Nicola J. Fawcett, Derrick W. Crook, Tim E. A. Peto, Martin J. Llewelyn, A. Sarah Walker.

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
