## [Decision Letter · Decision Letter 0]

26 Jul 2019

PONE-D-19-18063

Selective culture enrichment and sequencing of feces to enhance detection of antimicrobial resistance genes

PLOS ONE

Dear Dr. Peto,

Thank you for submitting your manuscript to PLOS ONE. After careful consideration, we feel that it has merit but does not fully meet PLOS ONE’s publication criteria as it currently stands. Therefore, we invite you to submit a revised version of the manuscript that addresses the points raised during the review process.

Your manuscript has been reviewed by three experts in your field. A major revision was needed before a final decided can be made.  

We would appreciate receiving your revised manuscript by 4 weeks. To enhance the reproducibility of your results, we recommend that if applicable you deposit your laboratory protocols in protocols.io, where a protocol can be assigned its own identifier (DOI) such that it can be cited independently in the future. For instructions see: http://journals.plos.org/plosone/s/submission-guidelines#loc-laboratory-protocols

We look forward to receiving your revised manuscript.

Kind regards,

Yung-Fu Chang

Academic Editor

PLOS ONE

Journal Requirements:

2. Please provide additional details regarding participant consent. In the ethics statement in the Methods and online submission information, please ensure that you have specified whether consent was suitably informed.

Reviewers' comments:

Reviewer's Responses to Questions

**Comments to the Author**

1. Is the manuscript technically sound, and do the data support the conclusions?

Reviewer #1: Yes

Reviewer #2: Yes

Reviewer #3: Yes

2. Has the statistical analysis been performed appropriately and rigorously? 

Reviewer #1: Yes

Reviewer #2: No

Reviewer #3: Yes

3. Have the authors made all data underlying the findings in their manuscript fully available?

Reviewer #1: Yes

Reviewer #2: Yes

Reviewer #3: Yes

4. Is the manuscript presented in an intelligible fashion and written in standard English?

Reviewer #1: Yes

Reviewer #2: No

Reviewer #3: Yes

5. Review Comments to the Author

Reviewer #1: Metagenomic analysis of antimicrobial resistance (AMR) genes in stool is an area of great interest and utility. Yet, it faces the problem of low detection of organisms present in low abundance (e.g., the Enterobacteriaceae). The authors propose a unique approach involving the selective enrichment of Enterobacteriaceae members before metagenomic analysis to increase detection efficiency. The also compare efficiency of two DNA extraction methods. Although, the results indicate variable success, the results should serve as a stepping stone for future investigations and improvements.

The article does satisfy the criteria listed for publication in PLOS ONE. The study presents the results of primary scientific research that has not been published elsewhere. The experiments are well conceived and well described. The rational for each experiment is adequately discussed. The results are appropriately presented. The conclusions are strongly based on data presented. Statistical analyses are appropriate and satisfactory. The writing style is very good and easy to follow. All ethical considerations are satisfactory. The article adheres to appropriate reporting guidelines and community standards for data availability.

Two minor typos were encountered

Line 172:

Sentence should not begin with a number i.e., “50μl”

Line 544:

“Effect of plasmid extraction of detection” should be “on detection”

Reviewer #2: This is a study focused on testing different methods for enrichment of Enterobacteriaceae for detection of carbapenemase gene and gyrA gene. The goal was to use an in vitro approach of spiking samples with the goal of developing new methodological areas that could be used as a focus for approaches using shotgun sequencing of samples focusing on the resistome (all antimicrobial resistance genes). This is an important area of research given current limitation of sequencing methods for detection of resistance genes in populations that although present in lower abundance, are of great relevance clinically. The methods seem adequate for the goals proposed, but lack enough information that would allow repetition of this approach. Given the strong methodological character of this study, this is of grave importance and need to be addressed. The title of the manuscript is misleading given the focus of the study, and should be re-phrased. The study does not present any statistical approaches, which should be added (at least what tools were used to compared descriptive results). The discussion should include more information about other studies focused on enrichment of samples for resistome analysis, and how outcomes of current method compare (e.g. “Enrichment allows identification of diverse, rare elements in metagenomic resistome-virulome sequencing . Noyes et al, 2017” - - as a disclosure, the reviewer is not involved with the research group from this manuscript)

Title – the title of this study is misleading given that it focused on specific bacteria (select Enterobacteriaceae) and specific resistance genes. I strongly suggest the title accurately reflect the conditions that were tested in the study.

Introduction

Line 54 – Could you further clarify what you mean with “binary measures of colonization”. It was not very clear to me what specifically you meant in relation to clonal dissemination of resistance.

Line 56-57 – I suggest adding that you mean “resistant” to antimicrobial drugs.

Line 62-64 – You talk about PCR here but make a reference to a microarray approach (reference number 7). I would suggest differentiating PCR (and maybe talking about qPCR instead) and microarray approach for detecting AMR genes.

Line 73 – I would suggest including Salmonella in the list of important enterobacteriaceae.

Line 86-89 – Having a section of the conclusion in the introduction seems atypical. I would suggest including what your hypothesis for the study was instead and presenting the conclusion of the study in the conclusion section.

Method

Line 100 – describe the methods used to label samples as ESBL negative. Culture one or two E coli isolates? Was a phenotypic approach used? Also please provide reference.

Line 126 – How was the 5 MacFarland suspension measured?

Line 128 – What nutrient broth was used?

Line 133- It is not clear what you meant “to allow quantification after sequencing” S. aureus was added to samples.

Line 139- Where was this approach based on? Any references for using this approach (e.g. specific antibiotics used, time of growth (why 8 hours?))? How would you compensate for difference concentration of bacteria affecting higher bacteria growth, given that bacteria have an exponential growth (not a linear growth)?

Line 146- Why were two different DNA extraction kits used, and why different enrichments used (one for each type of DNA extraction kit)?

Line 182- Although you mention different sources you used to annotate the dataset (metagenome, gyrA, plasmids), I did not see in this section what database you used to characterize resistance genes in the samples (e.g. beta-lactam resistance genes).

Line 196 – I understand the goal of the RDE approach, but the formula is not clear to me, and more information on what each variable means may resolve some questions. Also, where are the “4 unreached samples” coming from?

Statistical analysis

- Where is the statistical analysis section of this manuscript?

Results

Line 213-225- Part of this should be in the introduction and methods, not in the results. I suggest you be direct when presenting the results.

Line 230-241 – This should be in the methods. It is very confusing reading the enrichment section in the material in methods in part because this information is not present in that section.

Line 258 – Was this difference between unenriched and enriched significantly different? Where are the statistical methods used for this study?

Line 280 – What do you mean with “relative abundance of the spike organism of >= 90%? Relative abundance could indicate what percent of the bacteria are present in relation to the total population. Here do you mean that, or what % of the total bacteria amount spiked in the sample you are retrieving?

Figure 2 – Enrichment does seems to affect bacteria growth in a matter that is not linear as a greater amount of initial bacteria is spiked in the sample, as expected due to exponential growth of bacteria. How do you account for this effect on the quantification of resistance genes between samples with different initial amount of a bacteria of interest? ( in other words, how to you account for the inflation in the results caused by enrichment towards samples with higher initial AMR bacteria concentration)

Line 294 – By using Staphylococcus after enrichment, you accounted for the impact of all methods starting at and after DNA extraction on output between two or more different samples. However you did not account for the potential for unequal growth of a specific AMR bacteria caused by the enrichment process.

Line 303 – “outnumbered” – correct grammar error

Figure 5 – The legend of figure 5 does not seem to make sense ( I was able to understand what was happening based on the figure description ( the solid and dashed lines)

Discussion

Line 360-372 – None of the findings of the study are discussed in this paragraph and seems a repetition of content from the introduction. I suggest either inserting and linking finding of the study to the content provided here or removing this sentence from the manuscript.

Line 380-382 – Provide more information about what incubation period was selected, and why it was relevant, and how does it compare to other approaches available in literature

Line 392-394 – I think this is relevant information, but it is not clearly outlined here. I suggest better explaining what you meant here. One option is to give an example for a specific bacteria used in the study.

Line 401 - please link this information to the figure where this data is presented.

Conclusion

This study did not look at all AMR genes, but was limited to a small select number of genes. I would suggest adjusting the conclusion of the study to reflect the specific conditions that were tested, which are still very relevant.

Reviewer #3: The authors of this article have developed a very interesting topic of research namely metagenomic sequencing of fecal DNA to characterise an individual’s intestinal resistome. Metagenomics is a method that, instead of sequencing individual genomes, collectively analyzes all DNA isolated from a specific sample, representing all microorganisms. It can provides information about which organisms are present in the sample and what metabolic processes are possible in the community. The Authors of this study aimed to develop a hybrid protocol to improve detection of resistance genes in Enterobacteriaceae by using a short period of culture enrichment prior to sequencing of DNA extracted directly from the enriched sample. They chose culture conditions that would amplify a resistant subpopulation of Enterobacteriaceae of particular clinical importance, specifically those resistant to third generation cephalosporins, with the aim of bringing their resistance genes above the threshold of detection. Performance of the enrichment-sequencing protocol was assessed by spiking fecal samples with known concentrations of CPE. Their finding of such large variation in growth between different strains of Enterobacteriaceae precludes accurate estimation of their starting concentration unless their growth characteristics are known, undermining one of the aims of the assay. Another aspect of this protocol was the use of plasmid DNA extraction, which increased the detection of plasmid-mediated resistance in Enterobacteriaceae in preliminary spiking experiments. It was problematically that, plasmid DNA extraction produced unwanted artefacts because of differential extraction efficiency depending on plasmid size. In the conclusions, the authors stated that, alternative ways of quantifying scarce AMR genes are needed. However, their study demonstrates the need for any method to be validated in a range of well characterized conditions before it can reliably be used to make quantitative comparisons.

In the assessment of this manuscript I state that the concept of the presented research was well planned. In the "Abstract" section, the general assumption of the research undertaken is clearly presented. However, little information was given about the methodology used. Perhaps this is due to the limitations of the text volume in this section. The introduction is short, but it introduces the reader to the research topic in a sufficient way. The "Material and Methods" and "Results" sections present the subsequent stages of the research and the results in a detailed manner. In the "Discussion" section, the authors should compare the results of their own research with the results of other authors.

In summary, I find that the article is written in a way that meets the criteria of PLOS ONE. The applied research methods are well-chosen. In connection with this assessment, I recommend this manuscript for publication in the Journal PLOS ONE.

6. PLOS authors have the option to publish the peer review history of their article (what does this mean?). If published, this will include your full peer review and any attached files.

Reviewer #1: Yes: Samer Swedan

Reviewer #2: No

Reviewer #3: No

---

## [Author Response · Author response to Decision Letter 0]

20 Aug 2019

Response to reviewers comments (***identical to uploaded Response to Reviewers***)

We apologise that one correction was required to the data cited in line 352 of the revised manuscript. ‘E. cloacae was detected at a mean 12 RPKM (range 5-17)’ has been corrected to ‘E. cloacae was detected at a mean 8 RPKM (range 4-12)’. The incorrect numbers in the text were from a previous method of mapping reads to AMR genes. The mapping method described in the methods section had otherwise been used throughout, including in figures. This correction does not affect the interpretation or discussion.

Journal Requirements:

Headings reformatted

2. Please provide additional details regarding participant consent. In the ethics statement in the Methods and online submission information, please ensure that you have specified whether consent was suitably informed.

Informed consent added (new line 147-148)

Reviewer #1: Metagenomic analysis of antimicrobial resistance (AMR) genes in stool is an area of great interest and utility. Yet, it faces the problem of low detection of organisms present in low abundance (e.g., the Enterobacteriaceae). The authors propose a unique approach involving the selective enrichment of Enterobacteriaceae members before metagenomic analysis to increase detection efficiency. The also compare efficiency of two DNA extraction methods. Although, the results indicate variable success, the results should serve as a stepping stone for future investigations and improvements.

The article does satisfy the criteria listed for publication in PLOS ONE. The study presents the results of primary scientific research that has not been published elsewhere. The experiments are well conceived and well described. The rational for each experiment is adequately discussed. The results are appropriately presented. The conclusions are strongly based on data presented. Statistical analyses are appropriate and satisfactory. The writing style is very good and easy to follow. All ethical considerations are satisfactory. The article adheres to appropriate reporting guidelines and community standards for data availability.

Line 172:

Sentence should not begin with a number i.e., “50μl”

Corrected (new line 221)

Line 544:

“Effect of plasmid extraction of detection” should be “on detection”

Corrected (new line 585)

Reviewer #2: This is a study focused on testing different methods for enrichment of Enterobacteriaceae for detection of carbapenemase gene and gyrA gene. The goal was to use an in vitro approach of spiking samples with the goal of developing new methodological areas that could be used as a focus for approaches using shotgun sequencing of samples focusing on the resistome (all antimicrobial resistance genes). This is an important area of research given current limitation of sequencing methods for detection of resistance genes in populations that although present in lower abundance, are of great relevance clinically. The methods seem adequate for the goals proposed, but lack enough information that would allow repetition of this approach. Given the strong methodological character of this study, this is of grave importance and need to be addressed. 

Methods have been expanded, as outlined below

The title of the manuscript is misleading given the focus of the study, and should be re-phrased. 

Amended, as outlined below

The study does not present any statistical approaches, which should be added (at least what tools were used to compared descriptive results). 

Added, as outlined below

The discussion should include more information about other studies focused on enrichment of samples for resistome analysis, and how outcomes of current method compare (e.g. “Enrichment allows identification of diverse, rare elements in metagenomic resistome-virulome sequencing . Noyes et al, 2017” - - as a disclosure, the reviewer is not involved with the research group from this manuscript)

We appreciate being made aware of the study by Noyes 2017. We had mentioned another study (Lanza 2018) that also used target capture, but have now expanded this part of the discussion and refer to both studies. We also now refer to Raymond 2019, which used culture-enrichment and sequencing for the characterisation of fecal resistomes, but which was published after the original manuscript was drafted. New lines 389-392 & 454-466

Title – the title of this study is misleading given that it focused on specific bacteria (select Enterobacteriaceae) and specific resistance genes. I strongly suggest the title accurately reflect the conditions that were tested in the study.

The original title reflects the fact that this approach could apply to detection of resistance genes in any selectively culturable organisms, with third generation cephalosporin resistant Enterobacteriaceae chosen as a model. However, we see how this could be misinterpreted, and have amended the title to “Selective culture enrichment and sequencing of feces to enhance detection of antimicrobial resistance genes in third-generation cephalosporin resistant Enterobacteriaceae”

Introduction

Line 54 – Could you further clarify what you mean with “binary measures of colonization”. It was not very clear to me what specifically you meant in relation to clonal dissemination of resistance.

We have clarified that we meant classifying an individual as colonised or not with a resistant organism, versus measuring its abundance. New line 52

Line 56-57 – I suggest adding that you mean “resistant” to antimicrobial drugs.

Added. New lines 54-55

Line 62-64 – You talk about PCR here but make a reference to a microarray approach (reference number 7). I would suggest differentiating PCR (and maybe talking about qPCR instead) and microarray approach for detecting AMR genes.

Although the paper originally cited used PCR to detect AMR genes in feces, its main focus was validation of a microarray. We have cited two more relevant recent papers instead to make the original point about PCR. New lines 61-63

Line 73 – I would suggest including Salmonella in the list of important enterobacteriaceae.

Added. New line 72

Line 86-89 – Having a section of the conclusion in the introduction seems atypical. I would suggest including what your hypothesis for the study was instead and presenting the conclusion of the study in the conclusion section.

This was done to follow the PLOS ONE author instructions regarding the introduction (‘Conclude with a brief statement of the overall aim of the work and a comment about whether that aim was achieved’). We have therefore not changed it.

Method

Line 100 – describe the methods used to label samples as ESBL negative. Culture one or two E coli isolates? Was a phenotypic approach used? Also please provide reference.

Details added. New lines 153-156

Line 126 – How was the 5 MacFarland suspension measured?

Make of nephelometer added. New lines 183-184.

Line 128 – What nutrient broth was used?

Details added. New line 183.

Line 133- It is not clear what you meant “to allow quantification after sequencing” S. aureus was added to samples.

We have clarified that this was to allow estimation of the number of CPE organisms in post-enrichment samples. New lines 138-141.

Line 139- Where was this approach based on? Any references for using this approach (e.g. specific antibiotics used, time of growth (why 8 hours?))? How would you compensate for difference concentration of bacteria affecting higher bacteria growth, given that bacteria have an exponential growth (not a linear growth)?

The justification for the antibiotics used is given in lines 235-237 & 268-269 of the original manuscript (new lines 112-115 & 303-304). Selective culture in Mueller Hinton broth is commonly used in microbiological practice, and we consider that the conditions chosen are justified by the data presented in Figure 1. 

Exponential growth does not preclude quantification of the initial organism concentration, provided the doubling times are reasonably well known. This is demonstrated by the growth curves of the two different strains of E. coli in Figure 1, where during the exponential phase growth is linear on a log scale with the same slope. In fact, doubling times were different enough between different species of Enterobacteriaceae to cause a problem with quantification in the subsequent experiment, and this finding is emphasised in the discussion.

Line 146- Why were two different DNA extraction kits used, and why different enrichments used (one for each type of DNA extraction kit)?

Lines 255-262 of the original manuscript (new lines 302-305) describe why the failure of the initial approach of selective culture with vancomycin and metronidazole led to a revised protocol incorporating both more selective culture conditions (by adding cefpodoxime) and the use of an alternative extraction protocol (which we had shown in new lines 270-273 to have superior detection of plasmid-associated beta-lactamases).

We do not make any direct comparisons between results using the first and second enrichment protocols, so changing two aspects of the protocol simultaneously does not cause a problem in interpretation. As this point was not raised by other reviewers we have not made specific changes in response to this comment.

Line 182- Although you mention different sources you used to annotate the dataset (metagenome, gyrA, plasmids), I did not see in this section what database you used to characterize resistance genes in the samples (e.g. beta-lactam resistance genes).

We have clarified that we used genes from the assembled reference genomes of the spike organisms in the bioinformatics section of the methods. New lines 237-238.

Line 196 – I understand the goal of the RDE approach, but the formula is not clear to me, and more information on what each variable means may resolve some questions. Also, where are the “4 unreached samples” coming from?

We have clarified the different components of the formula, and clarified that the unenriched [not unreached] samples shown in Figure 3 were those that this relative detection efficiency (RDE) is calculated on. New lines 253-261.

Statistical analysis

- Where is the statistical analysis section of this manuscript?

The only statistical analysis was t-tests with unequal variance, which is now added to methods section (previously we had just referred to the test statistic when presenting the test results). New lines 263-265.

Line 213-225- Part of this should be in the introduction and methods, not in the results. I suggest you be direct when presenting the results.

The results were previously written to be understandable without reference to the methods. We agree that having the rationale for experiments in the methods is more in keeping with typical PLOS ONE style, and so we have restructured the methods and results sections. See new Experiment 1-4 sections of methods, new lines 91 -143. 

Line 230-241 – This should be in the methods. It is very confusing reading the enrichment section in the material in methods in part because this information is not present in that section.

We have re-ordered the presentation as suggested 

Line 258 – Was this difference between unenriched and enriched significantly different?

The difference is formally statistically significant (p<10-16¬) but we do not feel this is particularly helpful as we are not testing the hypothesis that enrichment increases the number of E. coli (this is known); instead we have added the ranges of mean relative abundance (which are completely non-overlapping, obviating the need for statistical testing) to the text instead. New lines 294-295.

Where are the statistical methods used for this study?

Added to the main Methods, as described above.

Line 280 – What do you mean with “relative abundance of the spike organism of >= 90%? Relative abundance could indicate what percent of the bacteria are present in relation to the total population. Here do you mean that, or what % of the total bacteria amount spiked in the sample you are retrieving?

We have clarified that this meant >90% relative abundance, as determined taxonomic classification of sequence data. New lines 308-309.

Figure 2 – Enrichment does seems to affect bacteria growth in a matter that is not linear as a greater amount of initial bacteria is spiked in the sample, as expected due to exponential growth of bacteria. How do you account for this effect on the quantification of resistance genes between samples with different initial amount of a bacteria of interest? ( in other words, how to you account for the inflation in the results caused by enrichment towards samples with higher initial AMR bacteria concentration)

We agree that these comparisons are not straightforward. First, exponential growth should be linear on a log scale. Figure 2 shows relative abundance, as this is the output from sequencing, but one would not expect this to be linearly related to the number of organisms present in the sample (e.g. if the number of organisms at 80% abundance doubled its new relative abundance would be 89%). This is why we used a standard spike of S. aureus to estimate the absolute number of organisms, and explain this rationale in the methods and results. That this provides more linearity is shown in Figure 3, but, as we show by comparing these estimates to culture in Figure 4, even this does not work adequately at high CPE concentrations. We consider reasons for this in the discussion in new lines 419-427. We have clarified this distinction between relative and absolute abundance in the results, new lines 307-313.

Line 294 – By using Staphylococcus after enrichment, you accounted for the impact of all methods starting at and after DNA extraction on output between two or more different samples. However you did not account for the potential for unequal growth of a specific AMR bacteria caused by the enrichment process.

We agree; for this method to quantify the starting concentration accurately growth must be sufficiently similar between Enterobacteriaceae. That we found this not to be the case is acknowledged in the abstract & discussion. New lines 34-36 & 404-417.

Line 303 – “outnumbered” – correct grammar error

Clarified. New lines 331-332.

Figure 5 – The legend of figure 5 does not seem to make sense ( I was able to understand what was happening based on the figure description ( the solid and dashed lines)

Clarified. New lines 359-365.

Discussion

Line 360-372 – None of the findings of the study are discussed in this paragraph and seems a repetition of content from the introduction. I suggest either inserting and linking finding of the study to the content provided here or removing this sentence from the manuscript.

As suggested, we have removed this paragraph re-iterating points made in the background, making the discussion more concise.

Line 380-382 – Provide more information about what incubation period was selected, and why it was relevant, and how does it compare to other approaches available in literature

Added information about how the incubation period was chosen as suggested. New lines 400-402. Comparison with other approaches is covered later in the Discussion.

Line 392-394 – I think this is relevant information, but it is not clearly outlined here. I suggest better explaining what you meant here. One option is to give an example for a specific bacteria used in the study.

We have clarified this point with reference to bacteria from the study. New lines 413-415.

Line 401 - please link this information to the figure where this data is presented.

Added. New line 423.

Conclusion

This study did not look at all AMR genes, but was limited to a small select number of genes. I would suggest adjusting the conclusion of the study to reflect the specific conditions that were tested, which are still very relevant.

We used these specific genes as positive controls in our experimental design – without knowing what genes are in a sample, it would not be possible to conduct the experiments we performed. We cannot see any reason why our findings with regard to differential doubling times and plasmid extraction would not be generalizable to other AMR genes in Enterobacteriaceae. We therefore consider that the conclusions as presented are reasonable and do not go beyond the data from this study - they only refer to AMR genes in Enterobacteriaceae, and they are applicable to selective enrichment using antibiotics other than the ones tested in the study. We note that neither of the other two reviewers had concerns about this.

Reviewer #3: The authors of this article have developed a very interesting topic of research namely metagenomic sequencing of fecal DNA to characterise an individual’s intestinal resistome. Metagenomics is a method that, instead of sequencing individual genomes, collectively analyzes all DNA isolated from a specific sample, representing all microorganisms. It can provides information about which organisms are present in the sample and what metabolic processes are possible in the community. The Authors of this study aimed to develop a hybrid protocol to improve detection of resistance genes in Enterobacteriaceae by using a short period of culture enrichment prior to sequencing of DNA extracted directly from the enriched sample. They chose culture conditions that would amplify a resistant subpopulation of Enterobacteriaceae of particular clinical importance, specifically those resistant to third generation cephalosporins, with the aim of bringing their resistance genes above the threshold of detection. Performance of the enrichment-sequencing protocol was assessed by spiking fecal samples with known concentrations of CPE. Their finding of such large variation in growth between different strains of Enterobacteriaceae precludes accurate estimation of their starting concentration unless their growth characteristics are known, undermining one of the aims of the assay. Another aspect of this protocol was the use of plasmid DNA extraction, which increased the detection of plasmid-mediated resistance in Enterobacteriaceae in preliminary spiking experiments. It was problematically that, plasmid DNA extraction produced unwanted artefacts because of differential extraction efficiency depending on plasmid size. In the conclusions, the authors stated that, alternative ways of quantifying scarce AMR genes are needed. However, their study demonstrates the need for any method to be validated in a range of well characterized conditions before it can reliably be used to make quantitative comparisons.

In the assessment of this manuscript I state that the concept of the presented research was well planned. In the "Abstract" section, the general assumption of the research undertaken is clearly presented. However, little information was given about the methodology used. Perhaps this is due to the limitations of the text volume in this section. The introduction is short, but it introduces the reader to the research topic in a sufficient way. The "Material and Methods" and "Results" sections present the subsequent stages of the research and the results in a detailed manner. In the "Discussion" section, the authors should compare the results of their own research with the results of other authors.

We have expanded the discussion to include more reference to other studies, as outlined above

In summary, I find that the article is written in a way that meets the criteria of PLOS ONE. The applied research methods are well-chosen. In connection with this assessment, I recommend this manuscript for publication in the Journal PLOS ONE.

---

## [Decision Letter · Decision Letter 1]

10 Sep 2019

Selective culture enrichment and sequencing of feces to enhance detection of antimicrobial resistance genes in third-generation cephalosporin resistant Enterobacteriaceae

PONE-D-19-18063R1

Dear Dr. Peto,

We are pleased to inform you that your manuscript has been judged scientifically suitable for publication and will be formally accepted for publication once it complies with all outstanding technical requirements.

With kind regards,

Yung-Fu Chang

Academic Editor

PLOS ONE

Additional Editor Comments (optional):

Reviewers' comments:

Reviewer's Responses to Questions

**Comments to the Author**

1. If the authors have adequately addressed your comments raised in a previous round of review and you feel that this manuscript is now acceptable for publication, you may indicate that here to bypass the “Comments to the Author” section, enter your conflict of interest statement in the “Confidential to Editor” section, and submit your "Accept" recommendation.

Reviewer #1: All comments have been addressed

Reviewer #2: All comments have been addressed

2. Is the manuscript technically sound, and do the data support the conclusions?

Reviewer #1: Yes

Reviewer #2: Yes

3. Has the statistical analysis been performed appropriately and rigorously? 

Reviewer #1: Yes

Reviewer #2: Yes

4. Have the authors made all data underlying the findings in their manuscript fully available?

Reviewer #1: Yes

Reviewer #2: Yes

5. Is the manuscript presented in an intelligible fashion and written in standard English?

Reviewer #1: Yes

Reviewer #2: Yes

6. Review Comments to the Author

Reviewer #1: I believe that the authors have satisfactorily addressed all points raised by the reviewers. The revised title more appropriately reflects the nature of the study. The abstract and the introduction are satisfactory. The description of the methods is improved and includes important details that are required for replication of experiments by others. The description of the results is improved and easy to follow. The discussion is greatly improved.

I have no additional comments.

Overall, the revised manuscript is suitable for publication in its current state.

Reviewer #2: I commend the authors for carefully considering each comment. All comments have been addressed by authors and I do not have any additional comments or suggestions.

7. PLOS authors have the option to publish the peer review history of their article (what does this mean?). If published, this will include your full peer review and any attached files.

Reviewer #1: No

Reviewer #2: No

---

## [Editor Report · Acceptance letter]

30 Oct 2019

PONE-D-19-18063R1 

Selective culture enrichment and sequencing of feces to enhance detection of antimicrobial resistance genes in third-generation cephalosporin resistant *Enterobacteriaceae*

Dear Dr. Peto:

I am pleased to inform you that your manuscript has been deemed suitable for publication in PLOS ONE. Congratulations! Your manuscript is now with our production department. 

With kind regards,

on behalf of

Dr. Yung-Fu Chang 

Academic Editor

PLOS ONE